# High Expression of NT5DC2 Is a Negative Prognostic Marker in Pulmonary Adenocarcinoma

**DOI:** 10.3390/cancers14061395

**Published:** 2022-03-09

**Authors:** Arik Bernard Schulze, Anna Kuntze, Lars Henning Schmidt, Michael Mohr, Alessandro Marra, Ludger Hillejan, Christian Schulz, Dennis Görlich, Wolfgang Hartmann, Annalen Bleckmann, Georg Evers

**Affiliations:** 1Department of Medicine A, Hematology, Oncology and Pulmonary Medicine, University Hospital Muenster, 48149 Muenster, Germany; michael.mohr@ukmuenster.de (M.M.); annalen.bleckmann@ukmuenster.de (A.B.); georg.evers@ukmuenster.de (G.E.); 2West German Cancer Center, University Hospital Muenster, 48149 Muenster, Germany; anna.kuntze@ukmuenster.de (A.K.); dennis.goerlich@ukmuenster.de (D.G.); wolfgang.hartmann@ukmuenster.de (W.H.); 3Gerhard Domagk Institute of Pathology, University Hospital Muenster, 48149 Muenster, Germany; 4Medical Department IV, Pulmonary Medicine and Thoracic Oncology, Klinikum Ingolstadt, 85049 Ingolstadt, Germany; larshenning.schmidt@klinikum-ingolstadt.de; 5Department of Internal Medicine II, University Hospital Regensburg, 93053 Regensburg, Germany; christian.schulz@ukr.de; 6Department of Thoracic Surgery, Rems-Murr-Klinikum Winnenden, 71364 Winnenden, Germany; alessandro.marra@rems-murr-kliniken.de; 7Department of Thoracic Surgery, Niels-Stensen-Kliniken, 48179 Ostercappeln, Germany; ludger.hillejan@niels-stensen-kliniken.de; 8Institute of Biostatistics and Clinical Research, Westfaelische-Wilhelms University Muenster, 48149 Muenster, Germany

**Keywords:** NSCLC, TP53, p53, NT5DC2, CAFs

## Abstract

**Simple Summary:**

Lung cancer is the leading cause of cancer related deaths and non-small cell lung cancer (NSCLC) is its most relevant subtype. Here, we present data on p53 co-playing 5′-Nucleotidase Domain-Containing Protein 2 (NT5DC2) protein- and gene-expression and its clinical implication and prognostic value. NT5DC2 overexpression was associated with advanced lymphonodal spread and reduced survival in patients with adenocarcinoma of the lung. Moreover, dysregulated p53 in combination with overexpressed NT5DC2 might promote malignancy by interaction with cancer associated fibroblasts.

**Abstract:**

Via immunohistochemistry (IHC) on tissue micro arrays (TMA) clinical and prognostic impact of p53 co-playing 5′-Nucleotidase Domain-Containing Protein 2 (NT5DC2) protein expression was evaluated in 252 NSCLC patients. Confirmatory, gene expression database. mRNA levels of NT5DC2 were studied in 1925 NSCLC patients. High protein expression of NT5DC2 resulted in reduced median overall survival (OS) of patients with stage I-III adenocarcinoma (ADC) (Log Rank *p* = 0.026, HR 2.04 (1.08–3.87)), but not in squamous cell carcinoma (SCC) (*p* = 0.514, HR 0.87 (0.57–1.33)). Findings on OS were reproduced via gene expression analysis in ADC (*p* < 0.001, HR 1.64 (1.30–2.08)) and SCC (*p* = 0.217, HR 0.86 (0.68–1.09)). Yet, NT5DC2 mRNA levels were higher in SCC compared to ADC (*p* < 0.001) and in pN2 tumors compared to pN0/1 tumors (*p* = 0.001). Likewise, NT5DC2 protein expression associated with high-grade SCC. Moreover, NT5DC2 expression was positively correlated with p53 protein (*p* = 0.018) and TP53 gene expression (*p* < 0.001) and its survival effect was p53 dependent. While p53 expression was negatively associated with the presence of CD34+ cancer associated fibroblasts (CAFs), NT5DC2 expression insignificantly tended to higher levels of SMA+ CAFs (*p* = 0.065).

## 1. Introduction

Globally, lung cancer is the most prevalent cause for cancer-related deaths [1]. For the analysis and understanding of the biological behavior and prognosis, it is important to distinguish between two different types of lung cancer. While small cell lung cancer (SCLC) is causal for about 14% of the cases, non-small cell lung cancer (NSCLC) comprises ≥ 85% of the lung cancer cases [2]. While SCLC is known for its high mutational load [3] and devastating prognosis, NSCLC includes targetable oncogenic driver mutations [4], such as Kirsten rat sarcoma (*KRAS*) [5], epidermal growth factor receptor (*EGFR*) [6,7], anaplastic lymphoma kinase (*ALK*) [8], Serine/threonine-protein kinase B-rapidly accelerated fibrosarcoma (*BRAF*) [9,10], rearranged during transfection (*RET*) [11], Proto-oncogene tyrosine-protein kinase ROS (*ROS1*) [12] and rarely neurotrophic tropomyosin receptor kinases (NTRK) [13]. Moreover, latest research has focused on co-mutations of multiple oncogenic and/ or tumor-suppressing pathways in NSCLC, as individual treatment approaches and prognostic implications might heavily differ [14,15,16]. While targeting oncogenic drivers is promising, addressing tumor suppressors in general is challenging and the clinical and therapeutical implications of co-mutations have not fully been understood [14]. Though in SCLC > 90% of the patients harbor Tumor Protein 53 (*TP53*) gene mutations [3], its mutational load in NSCLC with >25% is also frequently observed [17] and might even level to >30% in *ALK*, *ROS1* and *RET* mutated patients [14]. Due to loss of function (LoF), major tumor suppressor gene TP53 has so far not been therapeutically addressed in NSCLC but presence and absence of its protein p53 may sensitize to specific treatment strategies [18]. Therefore, co-players of up- and down-regulation of p53, such as 5′-Nucleotidase Domain-Containing Protein 2 (NT5DC2) may be targetable and are thus needed for biological and prognostic understanding of NSCLC. 

The latter protein NT5DC2 was shown to be upregulated in NSCLC cell culture. Its knockdown was associated with reduced A549 and H1299 cell culture growth in colony formation assays, as well as reduced migration and invasiveness in Transwell assays. In turn, overexpression of NT5DC2 enhanced proliferative abilities and eventually led to increased invasion and migration. However, lentiviral overexpression of NT5DC2 significantly reduced p53 messenger ribonucleic acid (RNA) levels and the effects of NT5DC2 knockdown by small interfering RNA (siRNA) were abrogated when p53 was simultaneously inactivated [19]. In addition, the impact of NT5DC2 on tumorigenesis and progression was also investigated in other entities. Here, knockdown of NT5DC2 resulted in limited progression of colorectal cancer cell models [20], whereas overexpression of NT5DC2 enhanced tumor growth in leiomyosarcoma [21], glioma [22] and hepatocellular carcinoma [23].

The aim of this work was to analyze the clinical and prognostic impact of cytoplasmic NT5DC2 protein level in NSCLC as well as its gene expression and a consecutive correlation with nuclear p53 protein expression and TP53 mRNA levels. To our knowledge, this is the first analysis to present clinical and prognostic data on NT5DC2 in two distinct NSCLC cohorts.

## 2. Materials and Methods

### 2.1. Study Collective

After admittance of the local ethics committee (Ref. Az 2016-445-f-S and Reg.Nr.: 4XMüller1), we retrospectively analyzed *n* = 252 patients that were diagnosed, and surgically treated for non-small cell lung cancer (NSCLC) between December 1998 and November 2004 at the Thoracic Surgery department of the St. Georg’s Clinic Ostercappeln for NT5DC2 protein expression. Date of surgical treatment determined the use of sixth edition of Tumor Nodule Metastasis (TNM) system proposed by the Union Internationale Contre le Cancer (UICC) [24]. An update towards later TNM staging system editions was not possible due to data protection and privacy concerns in respect with the ethical approval. Moreover, no fresh frozen tissue or separate tissue apart from tissue micro arrays was attainable. Hence, mutations and translocations in relevant locations such as *TP53-* [25], *EGFR*- [26], *ALK*- [27], *ROS1*- [28] and *BRAF*- [29] gene loci could not be assessed retrospectively. In *n* = 235 of these, we were able to perform correlation analyses on immunohistochemical p53 protein expression.

Moreover, we used the online messenger ribonucleic acid (mRNA) meta-analysis database kmplot.com (accessed on 3 March 2022) for NSCLC [30] to evaluate the prognostic effect of NT5DC2 (Affymetrix Probe ID 218051_s_at) and TP53 (Affymetrix Probe ID 201746_at) gene expression in *n* = 1925 patients.

### 2.2. Protein Expression Analysis and Immunohistochemistry

Primary tumor tissues, surgically resected at St. Georg’s Clinic Ostercappeln, were analyzed via 4-µm-thick formalin-fixed paraffin-embedded (FFPE) tissue microarrays (TMA). Each patient was represented by three punch cores from the original tumor specimen [31]. To perform immunohistochemistry (IHC), we used the peroxidase-conjugated avidin-biotin method. Primary antibodies contained rabbit polyclonal IgG anti-NT5DC2 antibody (Atlas Antibodies, SE-168 69 Bromma, Sweden, Catalog Number: HPA050683, Figure 1) and mouse monoclonal IgG_1_ anti-p53 (DO-7) antibody (Roche/Ventana, IN 46250-0457, Indianapolis, USA, Catalog Number: 790-2912, Appendix A). In brief, at ambient temperature we deparaffinized TMA sections in xylene and rehydrated tissue through graduated ethanol-solutions. Afterwards, the primary antibody was incubated for 30 minutes at ambient temperature. After washing, the incubation of the paraffin sections with biotinylated secondary antibodies was performed. For secondary antibody reaction, 3-amino-9-ethylcarbazole was used as a substrate and immunoreactions were detected (Roche/ Ventana OptiView DAB IHC Detection Kit, IN 46250-0457, Indianapolis, USA, Catalog Number: 760-700). Positive controls were performed on human prostate tissue for NT5DC2 (Figure 1F) and melanoma tissue for p53, respectively. Negative controls were found in human healthy lung tissue (Figure 1E). Tissue analysis was performed at an Olympus BX51 microscope with an average magnification of ×200 by AK, GE and ABS, respectively. Deviations in the microscopic analysis were discussed interdisciplinary. For each punch core we recorded the percentage tumoral, cytoplasmatic NT5DC2 and nuclear p53 stain (0–100%) and the staining intensity from 0-3, following Remmele et al. [32]. Analysis was performed by using the immunoreactive score (IRS) [33]. Here, an ordinal category of the percentage of positive tumor cells (i.e., 0 = no positive cells, 1 = ≤10% positive cells, 2 = 11–50% positive cells, 3 = 51–80% positive cells, 4 = >80% positive cells) is multiplied by the staining intensity (0, 1, 2, 3) (c.f., Figure 1 and Appendix A) and categorized as negative (0–1), mild positive (2–3), moderate positive (4–8) and strongly positive (9–12). For statistical analysis, negative and mild positive specimen (IRS ≤ 3) was summarized as ‘NT5DC2 low’/‘p53 low’ and moderate to strongly positive specimen (IRS 4–12) as ‘NT5DC2 high’/‘p53 high’.

### 2.3. Gene Expression Analysis

Regarding mRNA-based survival analysis, Kaplan-Meier plots were generated using kmplot.com (accessed on 3 March 2022). Based on lung cancer microarray mRNA analysis, data from Gene Expression Omnibus (GEO, ncbi.nlm.nig.gov/geo, ncbi.nlm.nih.gov/gds, accessed on 3 March 2022), Cancer Biomedical Informatics Grid (caBIG, biospecimens.cancer.gov/caBigTools.asp, accessed on 3 March 2022), and The Cancer Genome Atlas (TCGA, cancergenome.nih.gov, accessed on 3 March 2022) were merged to perform a meta-survival-analysis of NSCLC patients regarding the genes of interest [30]. To select probes for NT5DC2 expression, we chose the Affymetrix Probe ID 218051_s_at, resulting in the mentioned cohort sizes. For TP53 analysis, the Affymetrix Probe ID 201746_at was used. By dichotomizing low and high mRNA expression, resulting in a separation below and above the median mRNA expression of the gene set of interest [34], the Kaplan-Meier plotter tool divides the cohort. A correlation between IHC and mRNA expression could not be established as the generated data derive independent databases.

### 2.4. Post-Hoc Correlations with Cancer Associated Fibroblasts

The protein expression cohort has previously been analyzed for cluster of differentiation 34 (CD34) and alpha-smooth muscle actin (SMA) positive cancer associated fibroblasts (CAFs). Staining was carried out and analysis was performed as described previously [35]. Here, we used the frequencies of the detected CAFs to correlate with NT5DC2 and p53 protein expression.

### 2.5. Statistical Analysis

Cohort description was performed via raw count and frequencies, via mean and standard deviation (SD), and via median and interquartile range (IQR; Q1–Q3) or 95 percent confidence interval. Twofold associations between categorical variables were analyzed via Fisher’s exact test or Chi-square test, if applicable. Other than that, twofold associations of continuous and normally distributed variables were tested using unpaired *t*-test. Manifold associations of continuous and normally distributed variables were tested via One-way ANOVA. With respect to continuous and not-normally distributed variables or ordinal variables, associations were analyzed by Mann-Whitney-U test. Likewise, manifold associations of continuous and not-normally distributed or ordinal variables were evaluated via Kruskal-Wallis test.

Overall survival (OS) included the time (months) between histopathological diagnosis and censoring or death. Progression free survival (PFS) was defined as time (months) between histopathological diagnosis and censoring, death, first relapse or progress after initial treatment, depending on first chronological appearance. Univariate survival analyses compared OS and PFS between groups by using Log rank test. Kaplan Meier plots helped to visualize survival differences. Hazard ratios (HR) were univariately determined via Cox Regression using the inclusion method. Hazard ratios are presented with 95% Confidence interval (95% CI). Data collection as well as calculations were performed using IBM^®^ SPSS^®^ Statistics Version 27 (released 2020, IBM Corp., Armonk, NY, USA). Figure 1 and Figure 2 were fused in Adobe Photoshop Elements 2021, Version 19.0 (released 2021, Adobe Inc., San Jose, CA, USA). The Kaplan-Meier plotter online tool (kmplot.com, accessed on 3 March 2022) automatically analyzes data by using the statistical software R with its underlying ‘survplot’ command of the survival Bioconductor package [30,36]. We retracted the data via ‘export plot data as text’ and set up IBM^®^ SPSS^®^ Statistics Version 27 (released 2020, IBM Corp., Armonk, NY, USA) databases. The local significance level was set to 0.05. Due to the explorative character of the analysis, an adjustment to multiplicity was not determined.

## 3. Results

### 3.1. Protein Expression Cohort Analysis

Baseline characteristics of the present cohort analyzed via immunohistochemistry (IHC), consisting of *n* = 252 evaluable patients, can be found in Table 1. Patients were 65.9 years of age, 82% of the patients were male and 79% were smokers. Evaluation of preoperative performance status, assessed by Eastern Cooperative Oncology Group (ECOG) scoring, revealed good general condition (ECOG 0-I) in 92.5% of the patients. Moreover, regarding average forced expiratory volume in one second (FEV1 ≥ 80%), patients were not heavily obstructive.

While 48% harbored squamous cell tumor pattern, adenocarcinoma was present in 35% of the patients. Other than that, large cell histology was present in 17% of the patients. Pathologic grading resulted in predominantly high-grade (G3/G4) tumors (67.5%). Most patients were treated for stage I tumors (56%), but stage II (26%) and stage III (18%) disease were also present. As all tissue samples were from surgical treatment, metastastic tumors were not included in the protein expression analysis cohort (cM0 100%). Complete resection (R0) was achieved in 94% of the patients, while 5% had microscopic tumor infiltration in resection margin (R1) and 1% had macroscopic tumor infiltration (R2). Of all 252 patients, 53 patients received neoadjuvant radio-chemotherapeutic treatment. Moreover, nine patients obtained adjuvant chemotherapy and 42 patients underwent postoperative radiotherapy.

When considering low/high NT5DC2 protein expression, neither age, gender, performance status nor smoking habits were assigned to NT5DC2 IRS low (i.e., IRS score 0–3) or IRS high (i.e., IRS score 4–12) sub-cohorts (all *p* > 0.05). Likewise, histopathologic pattern (i.e., squamous cell carcinoma (SCC), adenocarcinoma (ADC), and large cell carcinoma (LCC)) was evenly distributed between NT5DC2 low and high subgroups (*p* = 0.534). However, LCC pattern was slightly more prominent in the NT5DC2 high cohort. Grading, pT, pN and pTNM staging did not differ significantly between the IRS low and high cohort (all *p* > 0.05). More so, when subdivided by histological pattern of SCC, ADC and LCC, NT5DC2 protein expression did not deduce TNM stage (i.e., Kruskal-Wallis *p* = 0.351, *p* = 0.119, and *p* = 0.483, respectively). Consistent with large cell pattern, high-grade tumors (i.e., G3/4 tumors) showed insignificant accumulation in NT5DC2 high sub-cohort.

Interestingly, when separated by high-grade (i.e., G3/4) and low-grade (i.e., G1/2) tumors, NT5DC2 overexpression was more prominent in high-grade diagnosed specimen (*p* = 0.029) in the overall protein expression cohort. While in ADC this grading dichotomy was not correlated with NT5DC2 expression (*p* = 1.000), in SCC however NT5DC2 protein overexpression significantly associated with G3/4 tumors (*p* = 0.003) (c.f., Appendix A).

In terms of progression-free survival and overall survival, the overall cohort did not differ significantly by IHC allocation of NT5DC2 (*p* = 0.144 and *p* = 0.146, respectively), but in both cases survival was insignificantly longer in the NT5DC2 low cohort.

When considering the stage dependent prognostic impact of NT5DC2 protein expression, there were no significant differences for disease stage I (*p* = 0.115), stage II (*p* = 0.739) and stage III (*p* = 0.986). Likewise, neither the absence of lymphonodal spread (pN0, *p* = 0.123) nor the presence of lymph node metastasis (pN1, *p* = 0.983; pN2, *p* = 0.746) resulted in favorable survival with respect to NT5DC2 protein expression.

However, if differentiated by histological pattern, high NT5DC2 expression in ADC resulted in a negative prognostic impact in terms of OS (c.f., Figure 2C, NT5DC2 low n.e. (95% CI n.e.) months vs. NT5DC2 high 53.2 (95% CI 28.3–78.2) months median OS, *p* = 0.026; HR 2.04 (95% CI: 1.08–3.87)). Yet, in SCC, NT5DC2 expression did not have prognostic impact (c.f., Figure 2B, NT5DC2 low 34.3 (95% CI 25.5–43.0) months vs. NT5DC2 high 35.6 (95% CI 28.1–43.1) months median OS, *p* = 0.514; HR 0.87 (95% CI: 0.57–1.33)).

After TMA analysis of immunohistochemical p53 staining, we performed correlation analyses regarding NT5DC2. Due to tissue loss, detection of necrosis or stroma bulks instead of tumor tissue, *n* = 17 patients could not be matched out of the original *n* = 252 patients. Therefore, the p53 correlations result in a reduced cohort of *n* = 235 patients.

First, we performed an Area under the curve Receiver-operating characteristic analysis (AUC ROC), positively correlating the IRS score (0–12) of p53 stain with NT5DC2 high tissue (*p* = 0.003, c.f., Appendix A). Moreover, low NT5DC2 expression (IRS 0–3) was associated with low p53 expression (IRS 0–3) (Table 1, *p* = 0.018). Additionally, an ordinate comparison of the IRS Score (negative = 0–1, mild = 2–3, moderate = 4–8, and strong = 9–12 stain) between nuclear p53 stain and cytoplasmatic NT5DC2 stain resulted in a positive correlation between both expressed proteins (c.f., Appendix A, *p* = 0.016). For the NT5DC2 low sub-cohort, mean p53 IRS was 2.1, while mean p53 IRS score in NT5DC2 high patients yielded 3.3 (*p* = 0.009). Similarly, median p53 expression was at an IRS score of 0 in the NT5DC2 low sub-cohort, while it increased to 1.8 in the NT5DC2 high sample analysis (*p* = 0.002).

### 3.2. Gene Expression Cohort Analysis

The gene expression cohort was extracted from the tool of kmplot.com (accessed on 3 March 2022), merging mRNA gene expression information from TCGA, caBIG and GEO. In total, we were able to evaluate *n* = 1925 patients here (c.f., Table 2). Statistical comparisons between protein and gene expression analysis cohort could not be performed, as these data derive from different patient cohorts. However, a descriptive comparison of the two cohorts is necessary to draw conclusions and confirm the hypotheses made. With respect to pT, pN and pM staging the gene expression analysis cohort harbors a comparable number of patients with pT1 (i.e., 37.9% mRNA vs. 27.4% IHC), pT2 (i.e., 51.1% mRNA vs. 58.3% IHC), pT3 (i.e., 7.0% mRNA vs. 10.3% IHC) and pT4 (i.e., 4.0% mRNA vs. 4.0% IHC) tumors, respectively. Likewise, lymphonodal spread, classified by pN category, is comparable (i.e., pN0 68.3% mRNA vs. 64.7% pN0 IHC, and pN+ 31.7% mRNA vs. 35.3% IHC, respectively). While the protein expression analysis cohort does not include metastasized patients, stage IV disease was diagnosed in 1% (0.4–1.4%) of in the gene expression cohort in a very small amount. In terms of pTNM staging, stage I is found in 64.5% (i.e., 55.6% in the protein expression cohort) of the cases and stage III is present in only 7.8% (i.e., 18.3% in the protein expression cohort) of patients. Both factors differ by about 10% from those in the protein expression analysis cohort.

Smoking habits are almost the same, with approximately 80% smokers and 20% non-smokers in both cohorts. However, with respect to histology, ADC is more prevalent in the gene expression cohort (57.8%) than in the protein expression cohort (35.3%) whereas SCC is less prevalent (i.e., 42.2% mRNA vs. 38.0% IHC). Moreover, female patients are more present in the gene expression analysis cohort (i.e., 39.4% mRNA vs. 18.3% IHC). Evaluation and comparison of tumor grading in the gene expression cohort is difficult as only 588 cases (30.5%) are evaluable. Regarding resection margins, in the kmplot.com (accessed on 3 March 2022) database, only R0 resected cases are selectable. Since we do not know the total number of ‘unknown’ vs. missing cases, we cannot conclude R1 and R2 resected patients, and therefore cannot compare with the protein expression cohort.

The exact comparisons of NT5DC2 low and high expression, as performed for histopathology using Fisher’s exact test in Table 3, are mostly not useful, as the dichotomy derives the median mRNA expression level in the specific sub-cohort testing. For this reason, statistical tests were not performed for gender, smoking status, pT stage, pN stage, pM stage, pTNM stage, grading and resection status.

In terms of survival parameters, the gene expression cohort shows significant survival differences in both OS (NT5DC2 low median OS 77.8 (95% CI 67.4–88.1) months vs. NT5DC2 high median OS 59.0 (95% CI 50.5–67.5) months, *p* = 0.0005; HR 1.25 (95% CI: 1.10–1.42), c.f. Figure 2E) and PFS (NT5DC2 low median OS n.e. (95% CI: n.e.) months vs. NT5DC2 high median OS 53.2 (95% CI: 36.1–70.2) months, *p* = 0.000046; HR 1.50 (95% CI: 1.23–1.81), c.f. Figure 2H) when differentiated by low and high NT5DC2 expression. Interestingly, when evaluated by histology, the ADC sub cohort shows a survival advantage with low NT5DC2 mRNA expression (i.e., NT5DC2 low 125.8 (95% CI: 101.9–149.7) months vs. NT5DC2 high 74.0 (95% CI: 56.7–91.3) months median OS, p = 0.000031; HR 1.64 (95% CI: 1.30–2.08), cf. Figure 2G), but the SCC sub cohort does not (i.e., NT5DC2 low 45.5 (95% CI: 30.0–60.9) months vs. NT5DC2 high 64.1 (95% CI: 44,5–83.7) months median OS, *p* = 0.217; HR 0.86 (95% CI: 0.68–1.09), cf. Figure 2F).

In parallel with analyses of the protein expression cohort, we extracted data on TP53 mRNA expression and NT5DC2 expression from kmplot.com (accessed on 3 March 2022) and merged them to assess correlation between both variables. Likewise, AUC ROC analysis associated higher mRNA TP53 values to NT5DC2 high tissue (c.f., Appendix A, *p* < 0.001). Here, mean mRNA expression count of TP53 was 423.2 in the NT5DC2 low cohort and mean mRNA expression count of TP53 increased to 745.4 in the NT5DC2 high cohort (Table 2, *p* < 0.001). As it were for mean values, median values show the positive correlation, too (*p* < 0.001).

### 3.3. NT5DC2 Protein and Gene Expression

With respect to quantitative NT5DC2 protein expression, immunohistochemical immunoreacitivity score (IRS, 0–12) was not associated with histological pattern. Here, for adenocarcinoma mean IRS was 4.49 and median IRS was 4.0. For squamous cell carcinoma the mean IRS was 3.95 and the median IRS was 4.0 (*p* = 0.221 and *p* = 0.328, respectively). However, for NT5DC2, quantitative mRNA levels in SCC were significantly higher (i.e., mean 641.7 and median 570.5 NT5DC2 mRNA count) than in ADC tissues (i.e., mean 478.3 and median 387.0 NT5DC2 mRNA count, *p* < 0.001 and *p* < 0.001, respectively). Otherwise, neither grading nor pT stage was significantly associated with NT5DC2 expression (c.f., Table 3). However, pN stage was significantly associated with NT5DC2 expression in the gene expression cohort (*p* = 0.001). Here, pN2 samples showed statistically higher mean (*p* = 0.001) and median mRNA count (*p* < 0.001) compared to pN0 and pN1 specimen. Accordingly, a non-significant trend in mean IRS score was also documented for the protein expression cohort (i.e., pN0 3.95, pN1 4.49, pN2 5.06; *p* = 0.114) (c.f., Table 3).

### 3.4. p53 Clinical Characterization in the Context of NT5DC2

In addition to the previously reported p53 expression, we focused on NT5DC2 expression in low and high p53 tumors. Here, we found a non-significant trend towards higher NT5DC2 protein expression in high p53 tumors (*t*-test *p* = 0.071 and Mann-Whitney U-test *p* = 0.073). However, in the gene expression analysis cohort, this correlation led to statistically significant results (c.f., Table 3, *t*-test *p* < 0.001 and Mann-Whitney-U test *p* < 0.001, respectively).

With additional focus on p53, a survival benefit was noted for SCC patients with elevated protein expression (i.e., *IRS strong* (9–12) median OS 95.8 (95% CI: 50.9–140.6) months vs. *IRS moderate* (4–8) median OS 35.3 (95% CI 32.1–38.5) months vs. *IRS mild* (2–3) median OS 54 (95% CI: 12.5–95.5) months vs. *IRS negative* (0–1) 26.2 (95% CI: 18.0–34.4) months, Log Rank test *p* = 0.025, c.f., Appendix A). Nevertheless, no significant differences in survival were observed with respect to the p53 staining in the ADC (i.e., *IRS strong* median OS 40.7 (95% CI: n.e.) months vs. *IRS moderate* median OS 44.1 (95% CI: 12.2–76.1) months vs. *IRS mild* median OS 42.6 (95% CI: 32.7–53.4) months vs. *IRS negative* median OS n.e., *p* = 0.237, c.f., Appendix A) or the overall cohort (i.e., *IRS strong* median OS 55.4 (95% CI: 12.2–98.6) months vs. *IRS moderate* median OS 35.9 (95% CI: 28.1–43.6) months vs. *IRS mild* median OS 38.5 (95% CI: 21.2–55.9) months vs. *IRS negative* median OS 46.3 (95% CI: 26.9–65.7) months, *p* = 0.800, c.f., Appendix A).

Other than that, the prognostic impact of NT5DC2 expression was highly contingent on p53/TP53 expression. While in SCC, survival of NT5DC2 patients did not differ, in ADC patients p53 protein expression altered the impact of NT5DC2 on overall survival significantly. Here, in p53 IRS low patients, NT5DC2 protein overexpression had a negative impact on survival (HR 2.35, 95% CI (1.12–4.97)), but in p53 IRS high patients, it was vice versa (HR 0.19, 95% CI (0.04–0.86), c.f., Table 4A). Likewise, in the gene expression cohort (c.f., Table 4B) patients with ADC and low TP53 mRNA levels had a reduced survival probability when overexpressing NT5DC2 (HR 1.86, 95% CI (1.31–2.64)). However, the negative prognostic impact of NT5DC2 was diminished, if TP53 expression was above the median mRNA level (HR 1.35, 95%CI (0.94–1.93)).

### 3.5. Post-Hoc Correlation of NT5DC2 and p53 Protein Expression with Cancer Associated Fibroblasts (CAFs)

Previously, we depicted the role of cluster of differentiation 34 (CD34)-positive CAFs as well as alpha-smooth muscle actin (SMA)-positive CAFs in the protein expression cohort [35]. Hence, we were able to perform post-hoc correlations of the detected frequencies of CAFs with tumoral p53 and NT5DC2 protein expression (c.f., Figure 3).

While NT5DC2 high tumors harbored insignificantly more SMA-positive CAFs (i.e., mean 15.0 (±SD 12.2)% and median 13.3 (IQR: 6.6–23.3)% SMA-positive of all stromal cells) than NT5DC2 low tumors (i.e., mean 12.2 (±SD 10.9)% and median 10.0 (IQR: 0.8–20.0)% SMA-positive of all stromal cells, *t*-test *p* = 0.065, Mann-Whitney U test *p* = 0.074), p53 expression had a negative significant correlation with CD34-positive CAFs. Here, tissue with low p53 expression was associated with a higher count of CD34-positive CAFs (i.e., mean 8.6 (±SD 17.4)% CD34-positive of all stromal cells and median 0.0 (IQR: 0.0–10.0)% CD34-positive of all stromal cells) compared to p53-protein enriched tissue (i.e., mean 3.2 (±SD 7.1)% CD34-positive of all stromal cells and median 0.0 (IQR: 0.0–3.3)% CD34-positive of all stromal cells, *t*-test *p* = 0.001, Mann-Whitney U test *p* = 0.030). There was no correlation between tumoral cytoplasmatic NT5DC2 expression and CD34-positive CAFs (*t*-test *p* = 0.630, Mann-Whitney U test *p* = 0.223) and between tumoral nuclear p53-expression and SMA-positive CAFs (*t*-test *p* = 0.650, Mann-Whitney U test *p* = 0.258).

When evaluating the cause of CD34-CAF association with p53, we previously stated, that CD34+ CAFs were primarily found in ADC patients (i.e., mean 14.3 (± 22.1)% of CAFs) but not in SCC (i.e., mean 2.9 (±5.5)% of CAFs) or LCC (i.e., mean 2.6 (±5.1)% of CAFs) (One way ANOVA *p* < 0.001) [37] and likewise p53 IRS was higher in SCC (i.e., mean 3.8 (±3.7) IRS score) than in ADC (i.e., mean 1.5 (±2.5) IRS score) or LCC (i.e., mean 3.0 (±3.5) IRS score) (One way ANOVA *p* < 0.001). However, if separated for histology, especially in ADC-patients the p53-expression alters the amount of invading CD34+ CAFs, resulting in 15.9 (±23.6)% of CAFs in p53 low ADC tumors and 6.7 (±12.9)% of CAFs in p53 high tumors (*p* = 0.038). This effect does not appear for SCC (*p* = 0.710) or LCC (*p* = 0.458).

Regarding tumor grading, neither in low-grade (i.e., G1/2) nor in high-grade (i.e., G3/4) tissue, NT5DC2 expression deduced SMA-positivity (i.e., ≥20% SMA+ of all stromal cells [35]) (*p* = 0.823, and *p* = 0.389, respectively) or CD34-positivity (i.e., ≥1% CD34+ of all stromal cells [35]) (*p* = 0.173, and *p* = 0.508, respectively) of stromal cells. Moreover, neither for SCC, nor for ADC growth pattern, NT5DC2 protein expression resulted in accumulation of SMA+ CAFs (i.e., *p* = 1.000, and *p* = 0.172, respectively). Likewise, CD34+ CAFs were not prone to NT5DC2 high tumors of SCC sub-type (*p* = 0.702) or ADC sub-type (*p* = 0.198), too.

As the gene expression cohort solely includes data of tumor-cell expressed factors, a correlation of the analyzed NT5DC2 and p53 with cancer associated fibroblast presence and CD34 and SMA expression of such was not possible.

## 4. Discussion

To our knowledge, this is the first study to present clinical and prognostic data on NT5DC2 in two separate NSCLC cohorts. In addition to Jin et al. [19], we are now able to add retrospective clinical data of protein and gene expression of NT5DC2 in NSCLC to preclinical in vitro data. Here, we found significant overexpression of NT5DC2 mRNA in pN2 lymphonodal positive tumors (c.f., Table 3), suggesting promotion of cell migration and invasiveness in vivo. Because the cohorts did not include sufficient data on distant metastasis, we were unable to correlate NT5DC2 expression with pM status. Yet, pT stage, which indicates the size of the primary tumor and its local invasiveness, did not result in different NT5DC2 protein and gene expression.

Nevertheless, we were able to translate the preclinical data of enhanced cell growth, migration, and invasiveness of NT5DC2 overexpressing NSCLC cell lines to the clinical setting. We demonstrated that ADC patients with high immunohistochemical cytoplasmatic NT5DC2 protein expression had lower overall survival compared to patients with low NT5DC2 expression (*n* = 89, *p* = 0.026). These findings were confirmed in the analyzed gene expression cohort, showing a significant survival benefit for low NT5DC2 mRNA expression (*n* = 719, *p* = 0.0005). However, for SCC patients, both in the IHC- and mRNA-analyzed cohort, this prognostic impact was not present (i.e., *p* = 0.514 for IHC and *p* = 0.217 for mRNA, respectively). 

While the overall survival analysis and the progression free survival analysis revealed significant differences in the gene expression cohort, this was not reproducible in the IHC-analyzed cohort (c.f., Figure 2). This effect might be influenced by the cohort’s distribution of the histological entities of ADC and SCC and the above-mentioned effect. ADC is overrepresented in the gene expression cohort (i.e., SCC 42.2%, ADC 57.8%, Table 2), but the histopathological frequency of ADC and SCC are reversely distributed in the protein expression cohort (i.e., SCC 48.0%, ADC 35.3%, LCC 16.7%, Table 1). 

Regarding the differences of SCC and ADC, Jin et al. showed that NT5DC2 protein and mRNA folds were comparable between A549 adeno NSCLC cells and H1299 squamous cells [19]. In contrast, we found a significant difference in the mRNA levels of NT5DC2 between SCC and ADC (c.f., Table 3). In the mRNA-analyzed cohort, SCC patients had higher levels (i.e., mean 641.7 folds vs. 478.3 folds, *p* < 0.001) than ADC patients. This histopathologic dichotomy needs to be further considered when studying NT5DC2 in NSCLC.

To further assess the impact of NT5DC2 on p53, we performed immunohistochemistry and correlated the IRS score of p53 with NT5DC2 protein expression. In contrast to cell culture analysis, where lentiviral NT5DC2 overexpression resulted in p53 mRNA downregulation [19], p53 expression was positively correlated with NT5DC2 expression in the protein expression cohort (*p* = 0.018). Moreover, semi-quantitative staining - as measured by IRS immunohistochemical expression—was significantly higher with respect to p53 mean (i.e., p53 IRS 2.1 in NT5DC2 low vs. p53 IRS 3.3 in NT5DC2 high, *p* = 0.009) and median expression (i.e., p53 IRS 0.0 in NT5DC2 low vs. p53 IRS 1.8 in NT5DC2 high, *p* = 0.002). These results were also reproduced in the gene expression cohort (i.e., NT5DC2 low: mean TP53 mRNA fold 423.2, median TP53 mRNA fold 303 vs. NT5DC2 high: mean TP53 mRNA fold 745.4, median TP53 mRNA fold 681.5, all *p* < 0.001). Similarly, AUC ROC analyses revealed a positive correlation of p53 and NT5DC2 (c.f., Appendix A).

To explain this difference, we must note that hetero- or homozygous TP53 tumor mutation is not known in any of the tissues studied. Unfortunately, we are not able to analyze genetic alterations in the protein expression cohort because fresh frozen samples are no longer available. Therefore, immunohistochemical p53 expression can only be detected as the presence of the addressed epitope of the counterstain of its protein and we cannot infer functional behavior of p53 at cellular level. In contrast, Jin et al. were able to knock down TP53 mRNA using small interfering RNA or lentivirally overexpress p53 [19]. To this point, the present retrospective data on mRNA levels and semi-quantitative immunohistochemical staining must be interpreted with caution.

However, in the protein expression cohort, strong p53 expression (IRS 9-12) in SCC patients had a favorable effect on outcome (c.f., Appendix A) whereas no statistically significant difference in survival was found for ADC patients (c.f., Appendix A) as well as in the overall cohort (c.f., Appendix A). In line with Jin et al. [19] we found the negative prognostic effect of protein and gene-overexpression of NT5DC2 in adenocarcinoma patients primarily in p53 low tissue. Other than that, elevated levels of p53 might mitigate (i.e., gene expression, c.f. Table 4B) or even reverse (i.e., protein expression, c.f., Table 4A) NT5DC2’s effect.

Moreover, p53 isoforms promote additional complexity. Here, the used primary p53 DO-7 antibody (Roche/Ventana, IN 46250-0457) only detects full length p53 epitopes. Nevertheless, previous studies have shown alternating p53 isoform effects. By lentiviral cloning of Δ133p53 plasmid in otherwise p53 deficient H1299 cells, Liu and Lin detected reduced chemosensitivity in Δ133p53 transfected cells compared to negative control plasmid transfected ones [37]. More so, via mouse model Roth et al. were able to depict a more invasive behavior of the Δ133p53 mouse analogue Δ122p53 carrying tumors [38]. Co-factors and modifiers, such as MDM-2 [39] have also not been evaluated here.

Basically, we can conclude, that NT5DC2 is p53/TP53 dependent. However, with respect to TP53 mutations and p53 isoforms, available TCGA-data on TP53 mutations in adeno-carcinoma patients (e.g., https://portal.gdc.cancer.gov/projects/TCGA-LUAD, accessed on 3 March 2022) cannot be matched to the small number of TCGA-derived gene-expression patients, as our ‘kmplot.com’ source does not allow for matching but are urgently needed for a better understanding of this complex interaction.

Zhu et al. hypothesized, that NT5DC2 activates tumor associated macrophages (TAMs) via vascular endothelial growth factor (VEGF) in a colorectal carcinoma cell culture model [20]. Unfortunately, we were unable to study TAMs in our protein expression cohort. However, with respect to the tumor microenvironment of colorectal cancer, it has been shown that cancer associated fibroblasts (CAFs) promote monocyte adhesion by secreting IL-8 and therefore enrich the tumor microenvironment with tumor-promoting M_2_ TAMs [40], and in parallel, influence vascular remodeling by upregulating VEGFa [41]. To this point, we post-hoc examined the protein expression cohort for cluster of differentiation 34 (CD34) and smooth muscle actin (SMA)-positive CAFs [35]. Here, previously the presence of CD34+ CAFs has been shown to be a marker of improved survival, and SMA+ CAFs were indicative for more advanced tumor stages [35]. Interestingly, p53 protein expression did not correlate with SMA+ CAFs, but strong nuclear p53 expression in tumor cells was negatively correlated with the presence of immuno-reactive CD34+ CAFs [42]. Vice versa, NT5DC2 expression did not show any correlation with the presence of CD34+ CAFs, but NT5DC2 protein overexpression resulted in an insignificant increase of SMA+ CAFs, that themselves are a marker of tumoral stromal remodeling [43] and more advanced tumor stages and lymphonodal spread in NSCLC [35].

Here, especially in adenocarcinoma, the p53 low, NT5DC2 low situation was associated with improved survival and likewise with higher CD34+ CAFs. In case of CD34-negative tumor microenvironment, the effect of NT5DC2 expression on survival was not apparent anymore (CD34+ environment: Log Rank *p* < 0.001, CD34- environment Log Rank *p* = 0.442). This too might allow for further investigation of NT5DC2 in combination with cancer associated fibroblasts.

Consequently, next research steps should include the comparative analysis between p53 wildtype A549 cells, p53 deficient H1299 cells and p53 mutant PC9 cells with respect to NT5DC2 overexpression and knockdown. Co-factors, such as MDM-2 expression must be considered. With further understanding, mammalian models might help to clarify the interactional potential towards tumor microenvironment and cancer associated fibroblasts.

## 5. Conclusions

In summary, by examining the protein expression, we hypothesized that high expression of NT5DC2 is a negative prognostic marker in the ADC subgroup of NSCLC patients and confirmed our hypothesis via gene expression analysis using the tool of kmplot.com (accessed on 3 March 2022). This finding was not evident in the SCC subgroup. Moreover, NT5DC2 was associated with lymphonodal spread (pN2 status) but not with pT-stage in the gene expression cohort. In addition, we observed a positive correlation of NT5DC2 and p53 protein and gene expression in both cohorts and depicted a p53 dependent NT5DC2 effect, warranting further investigation in future studies. Because NT5DC2 is a cytoplasmatic protein, which can be found not only in NSCLC but also in leiomyosarcoma [21], colorectal cancer [20] and hepatocellular carcinoma [23], therapeutic interactions are conceivable for multiple entities, and its interaction with p53 makes it a valuable target for further investigation. In particular, the impact of NT5DC2 with respect to p53 mutations, isoforms and deficiency itself and their effect on tumor stroma and on CAFs should further be investigated to determine whether it adversely affects the outcome of NSCLC course.

## Figures and Tables

**Figure 1 cancers-14-01395-f001:**
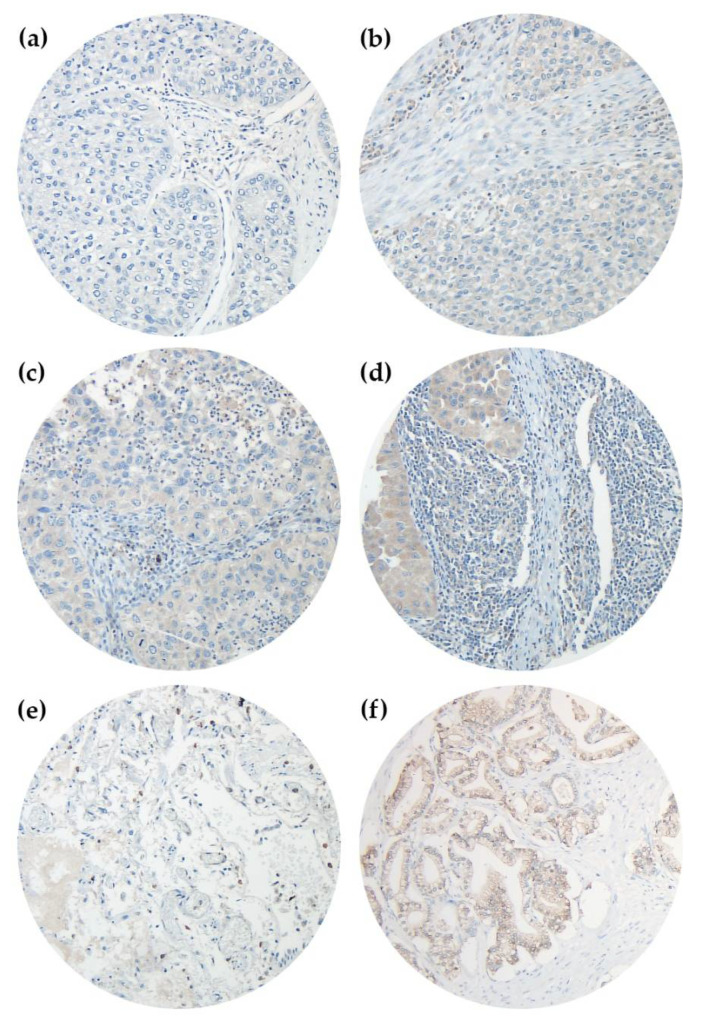
Immunohistochemistry of NT5DC2 in NSCLC TMAs. Primary antibody was Atlas Antibodies HPA050683 rabbit polyclonal IgG anti-NT5DC2 antibody, Secondary antibody was Roche/Ventana 760-700 OptiView DAB IHC Detection Kit. Analyses were performed on an Olympus BX51 microscope with an average magnification of ×200. (**a**) represents a negative core (IRS 0), (**b**) reveals an IRS intensity 1 core (IRS 1), (**c**) is an intensity 2 core (IRS 2) and (**d**) is an intensity 3 core (IRS 3), respectively. (**e**) represents healthy lung tissue, and (**f**) is positive control of human prostate tissue.

**Figure 2 cancers-14-01395-f002:**
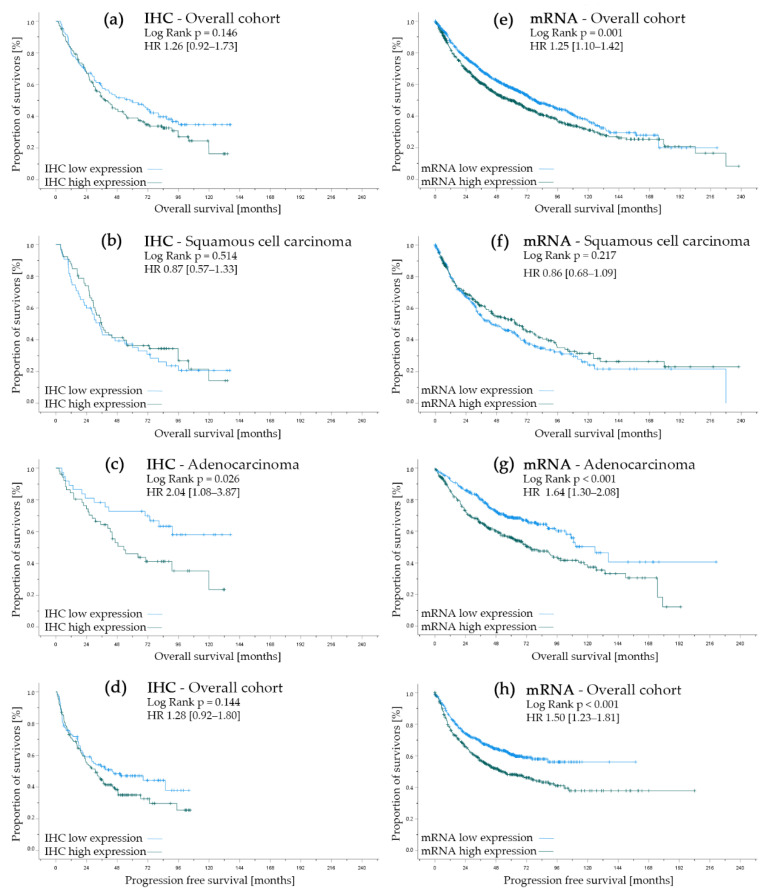
Univariate survival analysis regarding NT5DC2 expression. Left side (**a**–**d**) represents the protein expression analysis cohort where (**a**) documents the overall survival in the overall cohort, (**b**) represents the survival curves in squamous cell carcinoma, (**c**) displays the Kaplan–Meier curves in adenocarcinoma and (**d**) shows the progression-free survival of the overall cohort. Corresponding analyses can be found for the kmplot.com (accessed on 3 March 2022) gene expression cohort in (**e**) (overall survival in the overall cohort), (**f**) (overall survival in the squamous cell carcinoma sub-cohort), (**g**) (overall survival in the adenocarcinoma sub-cohort) and (**h**) (progression-free survival in the evaluable cohort). For (**a**–**d**), low expression is IRS 0–3, and high expression is IRS 4–12 (Atlas Antibodies, HPA050683). For (**e**–**h**), low expression is below median mRNA levels, and high expression is above median mRNA levels (Affymetrix ID 218051_s_at) for NT5DC2.

**Figure 3 cancers-14-01395-f003:**
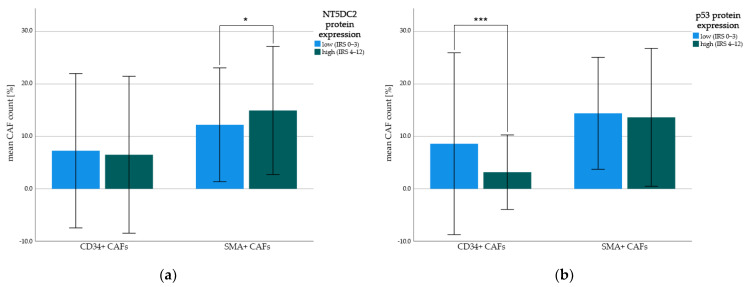
Mean count of CD34+ and SMA+ cancer-associated fibroblasts (CAFs). Protein expression was split by low (i.e., IRS 0–3) and high (i.e., IRS 4–12) immunohistochemistry staining intensity and correlated with mean frequency of CD34+ CAFs and SMA+ CAFs. Indexing * represent *t*-test *p*-values: (* *p* < 0.1, *** *p* ≤ 0.001). Error bars represent first standard deviation (±1SD). (**a**) NT5DC2 tumoral protein expression in correlation with CD34+ CAFs and SMA+ CAFs; (**b**) p53 tumoral protein expression in correlation with CD34+ CAFs and SMA+ CAFs.

**Table 1 cancers-14-01395-t001:** Baseline characteristics of the protein expression analysis cohort.

Variables		Total Cohort		NT5DC2 Low		NT5DC2 High		
		*n*_total_ = 252	% *	*n* = 107	% *	*n* = 145	% *	*p*-Value
Age (Years)	Mean (±SD)	65.9 (±8.5)		64.8 (±7.6)		66.8 (±9.0)		0.059 ^a^
Sex	male	206	81.7	86	80.4	120	58.3	0.625 ^c^
	female	46	18.3	21	19.6	25	17.2	
ECOG	0	33	13.1	16	15.0	17	11.7	0.253 ^b^
	I	200	79.4	85	79.4	115	79.3	
	II–III	19	7.5	6	5.6	13	9.0	
FEV_1_ (% pred.)	mean (±SD)	80.5 (±20.3)		80.6 (±20.8)		80.4 (±20.0)		0.950 a
	median (Q1–Q3)	81 (65–96)		82 (64–95.5)		81 (65.5–96.0)		0.999 b
Smoking status	non-smoker	54	21.5	20	18.7	34	23.6	0.438 ^c^
(* *n* = 251)	smoker	197	78.5	87	81.3	110	76.4	
Histopathology	squamous cell (SCC)	121	48.0	55	51.4	66	45.5	0.534 ^d^
	adeno (ADC)	89	35.3	37	34.6	52	35.9	
	large cell (LCC)	42	16.7	15	14.0	27	18.6	
p53 IHC	low (IRS 0–3)	151	64.3	70	73.7	81	57.9	**0.018 ^c^**
(* *n* = 235)	high (IRS 4–12)	84	35.7	25	26.3	59	42.1	
p53 IHC	mean (±SD)	2.8 (±3.4)		2.1 (±3.2)		3.3 (±3.5)		**0.009 ^a^**
(* *n* = 235)	median (Q1–Q3)	1.0 (0.0–5.0)		0.0 (0.0–4.0)		1.8 (0.0–6.0)		**0.002 ^b^**
Grade (* *n =* 249)	G1	1	0.4	1	0.9	0	0.0	0.080 ^b^
	G2	78	31.0	41	38.3	37	26.1	
	G3	135	53.6	50	46.7	85	59.9	
	G4	35	13.9	15	14.0	20	14.1	
Resection	R0	237	94.0	98	91.6	139	95.9	0.159 ^b^
	R1	13	5.2	8	7.5	5	3.4	
	R2	2	0.8	1	0.9	1	0.7	
UICC6 pT	pT1	163	27.4	36	33.6	33	22.8	0.251 ^b^
	pT2	55	58.3	54	50.5	93	64.1	
	pT3	34	10.3	13	12.1	13	9.0	
	pT4	10	4.0	4	3.7	6	4.1	
UICC6 pN	pN0	163	64.7	73	68.2	90	62.1	0.372 ^b^
	pN1	55	21.8	20	18.7	35	24.1	
	pN2	34	13.5	14	13.1	20	13.8	
UICC6 cM	cM0	252	100.0	107	100.0	145	100.0	1.000 ^b^
UICC6 pTNM	stage I	140	55.6	64	59.8	76	52.4	0.331 ^b^
	stage II	66	26.2	24	22.4	42	29.0	
	stage III	46	18.3	19	17.8	27	18.6	
PFS (months)	median (95% CI)	32.7 (22.4–42.9)		44.3 (9.2–79.4)		30.8 (22.0–39.5)		0.144 ^e^
OS (months)	median (95% CI)	43.9 (32.2–55.5)		58.1 (32.6–83.6)		38.6 (29.0–48.1)		0.146 ^e^
Follow up (months)	median (95% CI)	89.4 (84.1–94.7)		94.7 (86.8–102.6)		85.3 (82.5–88.1)		0.206 ^e^

* in % of non-missing values, indicated by italic n-values under variable; *p*-values: ^a^ student’s *t*-test, ^b^ Mann-Whitney-U Test, ^c^ Fisher’s exact test, ^d^ χ^2^ test, ^e^ Log Rank test; ECOG: Eastern Cooperative Oncology Group performance status, FEV1: forced expiratory volume in one second, IHC: immunohistochemistry, scored by immunoreactivity score (IRS), PFS: progression free survival, OS: overall survival. *p*-values in bold indicate significant differences.

**Table 2 cancers-14-01395-t002:** Baseline characteristics of the gene expression analysis cohort.

Variables		Total Cohort		NT5DC2 mRNA Low		NT5DC2 mRNA High		*p*-Value
		*n*_total_ = 1925	% *	*n* = 963	% *	*n* = 962	% *	
Sex (* *n =* 1814)	male	1100	60.6	550	60.6	550	60.7	#
	female	714	39.4	358	39.4	356	39.3	
	unknown	111		55		56		
Smoking status (* *n* = 1025)	non-smoker	205	20.0	102	19.9	103	20.1	#
	smoker	820	80.0	410	80.1	410	79.9	
	unknown	900		451		449		
Histopathology (* *n* = 1243)	squamous cell (SCC)	524	42.2	262	42.1	262	42.1	1.000 ^c^
	adeno (ADC)	719	57.8	360	57.9	359	57.9	
	unknown	682		341		341		
TP53 mRNA	low (<median expr.)	962	50.0	666	69.2	296	30.8	**<0.001 ^c^**
	high (≥median expr.)	963	50.0	297	30.8	666	69.2	
TP53 mRNA	mean (±SD)	584.2 (±428.4)		423.2 (±346.2)		745.4 (±442.1)		**<0.001 ^a^** **<0.001 ^b^**
	median (Q1–Q3)	488.0 (253.0–795.0)		303.0 (101.0–570.0)		681.5 (432.8–972.0)		
Grade (* *n* = 588)	G1	201	34.2	100	34.1	101	34.2	#
	G2	310	52.7	155	52.9	155	52.5	
	G3	77	13.1	38	13.0	39	23.2	
	unknown	1337		670		667		
Resection (* *n* = 726)	R0	726	100.0	363	100.0	363	100.0	-
	unknown	1199	0	600	0	599		
AJCC pT (* *n* = 1153)	pT1	437	37.9	218	37.9	219	37.9	#
	pT2	589	51.1	294	51.1	295	51.0	
	pT3	81	7.0	40	7.0	41	7.1	
	pT4	46	4.0	23	4.0	23	4.0	
	unknown	772		388		384		
AJCC pN (* *n* = 1144)	pN0	781	68.3	392	68.3	389	68.2	#
	pN1	252	22.0	126	22.0	126	22.1	
	pN2	111	9.7	56	9.8	55	9.6	
	unknown	781		389		392		
AJCC pM1 (* *n* = 691)	pM0	681	98.6	341	98.6	340	98.6	#
	pM1	10	1.4	5	1.4	5	1.4	
	unknown	1234		617		617		
AJCC pTNM (* *n* = 895)	stage I	577	64.5	288	64.4	289	64.5	#
	stage II	244	27.3	122	27.3	122	27.2	
	stage III	70	7.8	35	7.8	35	7.8	
	stage IV	4	0.4	2	0.4	2	0.4	
	unknown	1030		516		514		
PFS (months) (* *n* = 982)	median (95% CI)	88.7 (n.e.–n.e.)		n.e. (n.e.–n.e.)		53.2 (36.1–70.2)		<0.001 ^d^
OS (months)	median (95% CI)	69.0 (62.9–75.1)		77.8 (67.4–88.1)		59.0 (50.5–67.5)		<0.001 ^d^
Follow up (months)	median (95% CI)	69.4 (66.9–71.9)		69.0 (65.7–72.3)		71.8 (68.0–75.6)		0.081 ^d^

* in % of non-missing values, indicated by italic n-values under variable; in Histopathology unknown cases are not included into *p*-value calculation; *p*-values: # not calculated, ^a^ student’s *t*-test, ^b^ Mann-Whitney-U Test, ^c^ Fisher’s exact test, ^d^ Log Rank test; PFS: progression free survival, OS: overall survival. *p*-values in bold indicate significant differences. Unknown cases were depicted in italic absolute values.

**Table 3 cancers-14-01395-t003:** Mean and median NT5DC2 IHC IRS and mRNA count in both cohorts.

NT5DC2 Protein Expression	Mean (±SD) IHC IRS	*p*-Value			Median (Q1–Q3) IHC IRS	*p*-Value		
Overall	4.21 (±3.03)				4.0 (2.0–6.0)			
Histology								
SCC	3.95 (±2.88)	0.221 ^a^			4.0 (2.0–5.0)	0.328 ^b^		
ADC	4.49 (±3.37)				4.0 (2.0–8.0)			
UICC 6 pT				0.181 ^c^				0.122 ^d^
pT1	3.78 (±2.96)	0.073 ^a^			3.0 (2.0–6.0)	0.045 ^b^		
pT2	4.57 (±3.04)		0.157 ^a^		4.0 (2.0–6.0)		0.092 ^b^	
pT3	3.62 (±3.11)			0.988 ^a^	3.5 (1.8–4.0)			0.718 ^b^
pT4	3.60 (±2.63)				4.0 (1.5–6.0)			
UICC 6 pN				0.114 ^c^				0.345 ^d^
pN0	3.95 (±2.78)	0.259 ^a^			4.0 (2.0–6.0)	0.282 ^b^		
pN1	4.49 (±3.13)		0.467 ^a^		4.0 (2.0–8.0)		0.706 ^b^	
pN2	5.06 (±3.80)				4.0 (2.0–8.0)			
Grading				0.342 ^c^				0.259 ^d^
G1	1.00 (±n.e.)	0.333 ^a^			1.0 (1.0–1.0)	0.221 ^b^		
G2	3.79 (±2.85)		0.132 ^a^		3.0 (2.0–6.0)		0.140 ^b^	
G3	4.44 (±3.21)			0.607 ^a^	4.0 (2.0–6.0)			0.814 ^b^
G4	4.17 (±2.56)				4.0 (2.0–6.0)			
p53 IHC								
low (IRS 0–3)	4.02 (±3.13)	0.071 ^a^			4.0 (2.0–6.0)	0.073 ^b^		
high (IRS 4–12)	4.75 (±2.85)				4.0 (3.0–6.0)			
**NT5DC2** **Gene Expression**	**Mean (±SD)** **mRNA Count**	***p*-Value**			**Median (Q1–Q3)** **mRNA Count**	***p*-Value**		
Overall	636.2 (±454.6)				547.0 (330.0–835.0)			
Histology								
SCC	641.7 (±434.4)	<0.001 ^a^			570.5 (335.3–858.5)	<0.001 ^b^		
ADC	478.3 (±355.6)				387.0 (257.0–613.0)			
AJCC pT				0.568 ^c^				0.647 ^d^
pT1	706.0 (±431.6)	0.485 ^a^			635.0 (464.5–866.0)	0.981 ^b^		
pT2	725.0 (±427.9)		0.554 ^a^		621.0 (434.5–902.0)		0.829 ^b^	
pT3	698.5 (±370.7)			0.244 ^a^	652.0 (384.0–972.0)			0.321 ^b^
pT4	791.2 (±457.2)				705.5 (470.5–1071.3)			
AJCC pN				**0.001 ^c^**				**0.001 ^d^**
pN0	707.5 (±426.5)	0.354 ^a^			636.0 (449.0–870.0)	0.183 ^b^		
pN1	680.5 (±392.4)		**0.001 ^a^**		576.5 (407.5–886.3)		**<0.001 ^b^**	
pN2	848.9 (±436.4)				770.0 (528.0–1122.0)			
Grading				0.273 ^c^				0.821 ^d^
G1	844.8 (±580.2)	0.364 ^a^			736.0 (479.0–1066.0)	0.898 ^b^		
G2	802.9 (±374.9)		0.190 ^a^		722.0 (558.0–947.0)		0.521 ^b^	
G3	751.8 (±284.0)				695.0 (563.0–888.0)			
TP53 mRNA								
low (<median)	501.1 (±380.8)	**<0.001 ^a^**			390.5 (251.0–635.3)	**<0.001 ^b^**		
high (≥median)	771.2 (±481.5)				679.0 (484.0–952.0)			

IHC: immunohistochemistry, IRS: immunoreactivity score, mRNA: messenger ribonucleic acid, SCC: squamous cell carcinoma, ADC: adenocarcinoma; *p*-values: ^a^ student’s *t*-test, ^b^ Mann-Whitney-U Test, ^c^ One-way ANOVA, ^d^ Kruskal-Wallis test. *p*-values in bold indicate significant differences.

**Table 4 cancers-14-01395-t004:** Survival analysis in the protein expression cohort. NT5DC2 survival effect in Adenocarcinoma is highly dependent on p53 expression.

**(A) Protein Expression Cohort**						
**Histology**	**p53 Expression** **(IRS) ^1^**	**NT5DC2 Expression** **(IRS) ^1^**	**Median OS** (**95% CI) (Months)**	** *p* ** ** ^4^ **	**Hazard Ratio** **(95% CI)**	** *p* ** ** ^5^ **
SCC	Low	Low	23.8 (15.3–32.3)	0.402	*	0.403
		High	29.4 (17.1–41.7)		0.78 (0.44–1.39)	
	High	Low	43.9 (3.8–83.9)	0.466	*	0.467
		High	38.6 (15.6–61.5)		0.78 (0.39–1.54)	
ADC	Low	Low	n.e. (n.e.) ^3^	0.021	*	0.025
		High	48.7 (33.0–64.3)		2.35 (1.12–4.97)	
	High	Low	10.4 (5.8–15.0)	0.017	*	0.032
		High	90.9 (30.4–151.3)		0.19 (0.04–0.86)	
**(B) Gene Expression Cohort**						
**Histology**	**TP53 Expression** **(mRNA) ^2^**	**NT5DC2 Expression** **(mRNA) ^2^**	**Median OS** **(95% CI) (Months)**	** *p* ** ** ^4^ **	**Hazard Ratio** **(95% CI)**	** *p* ** ** ^5^ **
SCC	Low	Low	51.5 (31.5–71.6)	0.200	*	0.201
		High	62.2 (39.7–84.7)		0.78 (0.53–1.14)	
	High	Low	37.6 (23.0–52.2)	0.307	*	0.308
		High	64.1 (35.8–92.4)		0.84 (0.59–1.18)	
ADC	Low	Low	136.3 (n.e.) ^3^	<0.001	*	0.001
		High	63.0 (23.6–102.4)		1.86 (1.31–2.64)	
	High	Low	109.0 (101.6–116.4)	0.106	*	0.108
		High	75.4 (58.3–92.6)		1.35 (0.94–1.93)	

95% CI: 95% Confidence interval, SCC: Squamous cell carcinoma, ADC: Adenocarcinoma, ^1^ IRS low accounts for IRS 0–3, IRS high accounts for IRS 4–12, ^2^ mRNA low defined as <median mRNA count, mRNA high defined as ≥median mRNA count, ^3^ n.e. = not estimable, ^4^ Log Rank test, ^5^ Cox proportional hazards test, * NT5DC2 IRS/mRNA low was used as Index parameter for Hazard Ratio estimation.

## Data Availability

The data presented on protein expression analysis in this study are available on reasonable request from the corresponding author. These data are not publicly available due to ethical concern. The data presented on gene expression analysis are openly available in Győrffy, B., Surowiak, P., Budczies, J. and Lánczky, A. Online survival analysis software to assess the prognostic value of biomarkers using transcriptomic data in non-small-cell lung cancer. *PLoS ONE* **2013**, *8*, e82241. doi:10.1371/journal.pone.0082241 and website http://kmplot.com (accessed on 3 March 2022).

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
