# Peer review of "High Expression of NT5DC2 Is a Negative Prognostic Marker in Pulmonary Adenocarcinoma"

_cancers, 2022, doi:10.3390/cancers14061395_

Round 1

Reviewer 1 Report

The topic of the article is very interesting, since non-small cell lung cancer (NSCLC) comprises ≥ 85 % of the lung cancer cases.  The proposed idea to analyze p53 as well as  NT5DC2  is a reasonable step forward from current evidence implicating these biological functions in disease pathogenesis. However, the authors used only immunohistochemistry (IHC) data and gene expression database ‘kmplot.com’ analysis to study mRNA levels of NT5DC2 and TP53.

Authors should provide protein, mRNA analysis using Western blot and RT-PCR to investigate the expression of NT5DC2 and TP53 in biological samples. Immunohistochemistry is the quantitative method.

Reviewer 2 Report

Summary

This paper clearly shows that elevated levels of NT5DC2 at both mRNA and protein levels are associated with reduced survival in NSCLC patients with adenocarcinoma but not squamous-cell carcinoma, and that NT5DC2 that higher mean and median mRNA levels are elevated in pN2 tumours. Your observations that high levels of NT5DC2 are correlated with increased levels of p53 expression, and that high levels of p53 protein seemed to provide a favourable benefit to SCC but not ADC patients, are intriguing, but it is difficult to draw conclusions about the dichotomy between SCC and ADC with respect to the effects of elevated TP53, nor the fact that the relationship observed in this study between p53 and NT5DC2 is opposite to previous, in vitro research) given the lack of information on p53 mutational status (or isoform expression).However it does suggest this is an area worthy of further research.

General comments

Although your results concerning NT5DC2 are convincing, those regarding p53 are less clear and there is possibly a little too much emphasis on this in your summary. In addition, because of the observed differences between the SCC and ADC I would also like to see other analyses (eg staging and grading etc) performed on SCC and ADC individually. There were also sections where I found your argument difficult to follow and could possibly be clatified. Please refer to my comments and suggestions below.

Specific comments

Major items

In your analysis of association between NT5DC2 and tumour grade and stage I suggest that you repeat these analyses sub-cohorted into ADC and SCC given the different characteristics and prognoses for these two cancers that your other results suggest.  Similarly for the CAF analysis.

In paragraph lines 472-477 you state that the observation that strong expression of p53 had no effect on survival in ADC in your IHC patients could be explained by the interplay between NT5DC2 and p53. This paper showed that the reduced cell survival in cells treated with siNT5DC2 required p53, which would be consistent with your observation that low levels of NT5DC2 increased survival if TP53 carried a gain of function mutation, but does not explain why high TP53 expression (presumably driven by NT5DC2) would not be deleterious unless this was WT p53 that might act to mitigate against other effects of elevated NT5DC2. Conversely, strong expression of p53 as a positive prognostic indicator is consistent with Jin et al. Please consider rewriting this paragraph to clarify what you mean.

Regarding the differences between the ADC and SCC with respect to TP53 levels, it could also reflect different roles for alterations in these two cancers. As you acknowledge, it is extremely difficult to interpret these results without knowledge of p53 mutational status (This data can however, be obtained from at least some of the databases used in the mRNA analysis. See, for example, Zeng et al. 2021 doi: 10.1186/s12890-021-01671-8 )

In addition, there in increasing evidence that different isoforms of TP53 have different effects. The antibody used will only detect full-length TP53 but not ∆133, and will not discriminate between C-terminal isoforms, not does the probe used for the mRNA analysis. These matters could be addressed further in the discussion section, and I would also suggest that the summary and abstract etc be more focused on NT5DC2 rather than TP53.

Minor items

At lines 240, 243, 325, 327, 331 and 333, 439  it should refer to figure 1 not figure 2. Also, in the legend it would be helpful to mention which study population (your retrospective patient group or the combined database samples) are shown.  I am also assuming that figure 3 should actually be figure 2?

At line 301 should refer to table 1 not table 2

In Figure 1 please include a specific reference to which group is represented by which graphs (ie A, B, C, D show data from the IHC cohort etc)

Consider subdividing table 3 so that readers can see from the title heading which cohort you are presenting

At lines 200-201 and 223-224 you mention a small number of LCC patients. Did you include these in your overall analysis or omit them? Did you consider looking at these as a separate sub-cohort in your mRNA expression analysis?

Why do you think that elevated p53 protein expression should provide a survival benefit in SCC when this is also being associated with a reduction in cancer-associated fibroblasts, despite CD34+CAFs being a marker of improved survival?

Round 2

Reviewer 1 Report

The article could be published

Reviewer 2 Report

Thankyou for considering my suggestions and incorporating a number of changes. These have significantly clarified the manuscript.